# A Novel Physics-Statistical Coupled Paradigm for Retrieving Integrated Water Vapor Content Based on Artificial Intelligence

Ruyu Mei [1,2,†], Kebiao Mao [2,3,*,†], Jiancheng Shi [4], Jeffrey Nielson [5], Sayed M. Bateni [6], Fei Meng [1] and Guoming Du [7]

1 School of Surveying and Geo-Informatics, Shandong Jianzhu University, Jinan 250100, China; 2021160104@stu.sdjzu.edu.cn (R.M.); lzhmf@sdjzu.edu.cn (F.M.)
2 Institute of Agricultural Resources and Regional Planning, Chinese Academy of Agricultural Sciences, Beijing 100081, China
3 School of Physics and Electronic-Engineering, Ningxia University, Yinchuan 750021, China
4 National Space Science Center, Chinese Academy of Sciences, Beijing 100190, China; shijiancheng@nssc.ac.cn
5 Department of Watershed Sciences, Utah State University, Logan, UT 84322, USA; jeffrey.nielson@wsu.edu
6 Department of Civil and Environmental Engineering and Water Resources Research Center, University of Hawaii at Manoa, Honolulu, HI 96822, USA; smbateni@hawaii.edu
7 School of Public Administration and Law, Northeast Agricultural University, Harbin 150006, China; duguoming@neau.edu.cn
* Correspondence: maokebiao@caas.cn
† These authors contributed equally to this work.

**Abstract:** Retrieval of integrated water vapor content (WVC) from remote sensing data is often ill-posed because of insufficient observational information. There are many factors that cause WVC changes, which yield instability in the accuracy of many traditional algorithms. To overcome this problem, we developed a novel fully-coupled paradigm for the robust retrieval of WVC from thermal infrared remote sensing data. Through the derivation of the physical radiative transfer equation, we determined two conditions that need to be satisfied for the deep learning retrieval paradigm of WVC. The first condition is that the input parameters and output parameters of the deep learning need to be able to build a complete set of solvable equations in theory. The second condition is that, if there is a strong relationship between input parameters and output parameters, it can be directly retrieved. If it is a weak relationship, we need to use prior knowledge to improve the portability and accuracy of the algorithm. The training and test data of deep learning is composed of representative solutions of physical methods and solutions of statistical methods. The representative solutions of the physical methods were obtained from the physical forward model, and the statistical solutions were obtained from multi-source data which can compensate for the defect that the physical model cannot simulate mixed pixels. MODIS L1B data was used for case analysis of paradigm retrieval, and the analysis indicated that four thermal infrared bands were usually needed as the input parameters of deep learning and the integrated water vapor content as the output parameter. When land surface temperature and emissivity were taken as prior knowledge, the root-mean-square error (RMSE) of the retrieved WVC was 0.07 $g/cm^2$. The optimal accuracy RMSE was 0.27 $g/cm^2$. When there was a strong correlation between input parameters and output parameters, i.e., if there were two bands that were very sensitive to WVC in the band combination, high-precision retrieval could also be achieved without prior knowledge. All the analyses show that the paradigm of deep learning coupling physics and statistics can accurately retrieve WVC, which is a significant improvement on the traditional method and solves the problem of lack of physical interpretation of deep learning.

**Keywords:** integrated water vapor content; radiative transfer; deep learning; prior knowledge

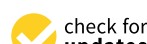

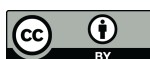

## 1. Introduction

Temporal and spatial variations of integrated water vapor content (WVC) influence the transfer of solar radiation, thus affecting many weather and climate change processes [1].

For example, variation in WVC affects global heat circulation, water circulation, and radiation energy balance [2]. WVC is the most influential variable in the atmosphere on solar radiation. In the process of radiative transfer, integrated water vapor content absorbs and releases surface long-wave radiation, and latent heat through phase transformation, which impacts air and ground temperatures. Retrieval of WVC plays an important role in drought monitoring [3,4], climate change [5], land surface temperature retrieval [6], and other studies.

Common methods of estimating WVC include radiosonde technology, the Global Navigation Satellite System (GNSS), and remote sensing techniques. Traditionally, the study of WVC has mainly relied on radiosonde technology [6], where WVC is determined through profiling measurements of temperature, relative humidity, and total pressure. However, radiosonde methods often have high operating costs, low sounding frequency, and large distances between launch points relative to the spatial variation of WVC [7], and therefore are only practical for small-scale use. In recent years, gradual improvements in the GNSS have increased its potential for WVC detection. A GNSS signal experiences tropospheric wet delay when it travels through the neutral atmosphere between satellite and receiver, and can achieve mm-scale accuracy [8–10]. This method is based on the upward detection of water vapor from the surface and is not affected by complex surface radiation and reflection information. Using GNSS, continuous WVC monitoring is possible from fixed stations on the earth's surface [11,12]. GNSS monitoring of WVC is not limited by weather [13] and is widely used in the evaluation of water vapor products [14–16]. Unfortunately, the distribution of detectors is often uneven and cannot accurately resolve gradient changes of water vapor on the horizontal scale, which limits its utility. Satellite remote sensing can quickly and widely obtain large-scale spatial information, which can improve the spatio–temporal analysis of WVC at regional scale [17]. Based on the absorption characteristics of water vapor, remote sensing uses visible-near infrared (VNIR) [18,19], thermal infrared (TIR) [20] bands and microwave bands [15,21,22] for the estimation of WVC. The rapid development of remote sensing technology has yielded an abundance of data and methods for retrieving WVC.

Statistical, physical, and machine learning methods are used to obtain WVC through remote sensing. Many statistical methods rely on the regression of brightness temperature (BT) and WVC. This has the advantage of simplicity, requiring only a few parameters, but the physical meaning is not very clear and the models are often not highly portable. Through the remote sensing method, we cannot accurately obtain all the parameters affecting WVC retrieval. In order to reduce this ill-posed problem and improve the accuracy, principal component regression [23,24], empirical regression [23], Bayes [25,26], and other methods have been proposed. The surface types in different regions are different, so the retrieval method obtained by using statistical methods is mainly suitable for local regions. Although the physical method is based on energy balance and has high retrieval accuracy [27,28], it requires many parameters. Due to the insufficient information for remote sensing observation, the number of unknowns is more than the number of equations, which forms an "ill-conditioned" problem. In order to reduce the unknowns, thermal infrared remote sensing usually used the split window algorithm (two thermal infrared bands) to retrieve the WVC [29–31], which was also affected by the surface type. The principle of this method is that, under the premise that integrated water vapor content has spatial invariance in a specific region [32], using the linear approximation of the Planck function to the brightness temperature difference, linear approximation is achieved of the relationship between brightness temperature difference and absorption of WVC. However, this requires that the WVC is not too high and that the land surface specific emissivity does not vary much between bands. To improve the accuracy of integrated water vapor content retrieval, Liu et al. [20] and Hu et al. [33] added a water vapor band to the traditional split window algorithm and proposed a three-band algorithm, which improved the retrieval accuracy and reduced the uncertainty of land surface emissivity under dry atmospheric conditions (i.e., WVC < 2 $g/cm^2$). Machine learning methods often use neural networks to obtain a

relationship between WVC and various input parameters [34–36]. The accuracy of these methods depends on the accuracy of the training and testing datasets. The advantage of machine learning is that it can handle nonlinear calculations to obtain an approximation of the optimal solution. However, it is difficult for these methods to give a specific explanation of physical relations between input variables and output parameters, so they are often regarded as "black boxes".

In order to overcome the shortcomings of traditional methods, this study constructs a retrieval paradigm of water vapor content based on deep learning (DL) coupled physical and statistical methods. Firstly, the basic conditions for determining the DL paradigm were derived based on physical methods, and then the representative solutions of the physical methods were obtained using forward physical models and the solutions of statistical methods were obtained using multi-source data. Finally, DL used the solutions of physical and statistical methods as training and test data to achieve the purpose of coupling, and analyzed whether the retrieval paradigm needs to use land surface temperature (LST) and land surface emissivity (LSE) as prior knowledge to improve the retrieval accuracy and algorithm portability.

## 2. Methodology

The DL method is essentially the same as other methods utilized in order to improve the accuracy of target parameters. If we want to obtain a general inversion paradigm, we must prove that the output parameters of DL can be uniquely determined by the input parameters. To construct a WVC retrieval paradigm of DL with physical interpretation, we used the thermal radiation energy balance equation to determine the relationship between the input and output parameters of DL, and ensured that a complete set of retrieval equations can be constructed theoretically between the input and output parameters. The retrieval paradigm is shown in Figure 1. Firstly, based on the derivation of the energy radiation balance equation, it is theoretically determined how many equation sets (physical method) need to be constructed to solve the integrated water vapor content, that is, how many input parameters need to be determined for DL, and then used the forward model MODTRAN simulation to obtain the representative solution of the physical method. Secondly, the forward model simulates pure pixels, and representative solutions are needed for large-scale mixed pixels. Based on the derivation of the physical model, we built a fuzzy statistical method, and used the multi-source data to obtain a reliable statistical method solution as a supplement to the physical method solution. Thirdly, the solutions of physical and statistical methods formed the training and testing database of DL, thus realizing the purpose of coupling physical and statistical methods of DL. Fourthly, the information for the TIR window band under clear sky conditions is mainly affected by LST and LSE, and the retrieval accuracy of LST and LSE is high. Therefore, in order to improve the accuracy of WVC retrieval in the TIR window band and the portability of the algorithm, we can take the LST and LSE as a priori knowledge. Here we used two sets of models for comparative analysis, one of which added LST and LSE as prior knowledge, and the other group had no prior knowledge. Fifthly, we used site observation data for precision verification and cross validation with MODIS products.

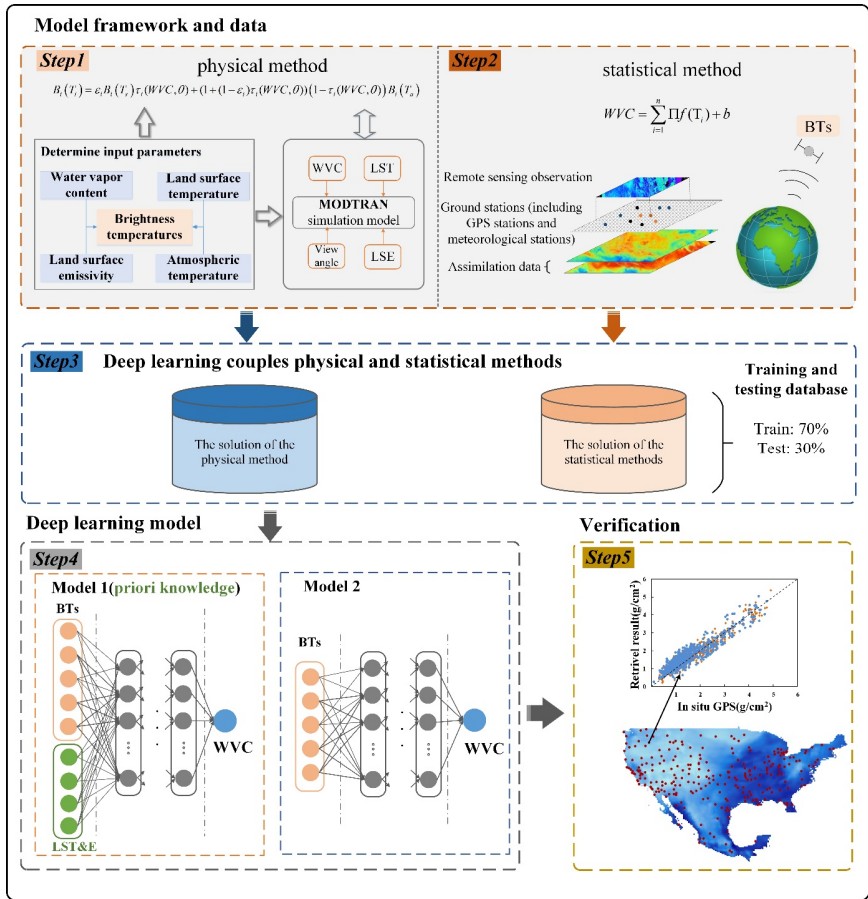

**Figure 1.** A novel physics-statistical coupled deep learning paradigm for retrieving integrated water vapor content.

## 3. Materials and Methods

### 3.1. Data

The retrieval paradigm of deep learning coupling physics and statistical methods proposed in this study is realized by composing the representative solutions of physical methods and statistical methods into a training and testing database, so the acquisition of both physical methods and statistical methods solutions (WVC) is the key. We used the classical atmospheric radiation transfer model MODTRAN to simulate the thermal radiation transfer in order to obtain the physical method solution, and used the multi-source data to obtain the statistical method solution, so as to make up for the defect that it is difficult to simulate the mixed pixel with the former model.

#### 3.1.1. Simulated Data

MODTRAN is a medium spectral resolution radiative transfer model, which is widely used and has high accuracy [37]. It can simulate the entire radiative transfer process, by setting a variety of land-surface, atmospheric, and instrumentation parameters. Through radiative transport simulations, every transport parameter in the forward process of a single thermal infrared band is known. For this study, atmospheric WVC is the target parameter of the physical method (the solution of the equation). The mid-latitude summer atmospheric reference model is selected as the input model of MODTRAN. For the required parameters, we used 17 surface types, including mainly soil, vegetation, water and rock. Other parameter settings are as follows: (1) surface temperature: the training data range is 280–325 K, with a step interval of 2 K; the test data range is 281–323 K, with a step interval of 3 K; (2) integrated water vapor content: the training data range is 0.2–4.5 g/cm$^2$, with a step interval of 0.3 g/cm$^2$; the test data range is 0.3–3.8 g/cm$^2$, and the step interval is

0.4 g/cm$^2$; (3) the observation angle range of the satellite ranges from 0 to 65°, with step intervals of 3°. Based on the different features of MODIS bands 27, 28, 29, 31, and 32, the model was set up and iteratively simulated to obtain the brightness temperature at satellite. The various parameters of the simulation process are known, so we can obtain solutions for physical methods under different conditions, which can form a part of the deep learning training and testing data. In addition, when the observation angle is relatively large, it is necessary to exclude some simulated values because the water vapor on the oblique path is already large (>5 g/cm$^2$) and there is not much information obtained through thermal infrared remote sensing.

### 3.1.2. Remote Sensing Data

MODIS remote sensing data was selected as case analysis data, which is a new generation of "union of imagery and spectrum" optical remote sensing instruments. The program consists of two polar-orbiting satellites of the EOS (Earth Observation System) in the United States, called 'Terra' and 'Aqua', and can provide day and night images of any point on the earth's surface every 1–2 days, with a spatial resolution of 1 km at nadir, and 5 km at the edge of the scan [38]. MODIS has lower instrument noise and narrower spectral response functions (SRF) [39], and can provide high quality data. The sensor has 36 bands (0.62–14.38 μm), with multiple TIR bands, so that multiple combination analysis can be performed. The MODIS sensor also has multiple TIR bands suitable for WVC retrieval, including band 27 (centered at 6.72 μm), which is located near the center of the water vapor absorption region, monitoring radiation from the upper atmosphere. Band 28 (centered at 7.33 μm) and band 29 (centered at 8.55 μm) are closer to the wing of the absorption zone, which enables the sensor to detect radiation from the lower layer [20]. Band 31 (centered at 11.03 μm) and band 32 (centered at 12.02 μm) are in the atmospheric window, which is mainly affected by water vapor content. Furthermore, MOD/MYD11 products include LST and LSE, and the error of LST is typically 1 K under stable atmospheric conditions, with LSE products under 0.02 [40]. Due to the different emissivity of different surface types, the TIR long-wave radiation is quite different [41]. Therefore, in order to obtain the surface type information, we took the band emissivity as prior knowledge. The MOD/MYD05 product contains WVC products retrieved by the VNIR ratio method and the TIR split window algorithm, and the accuracy of MOD/MYD05 has been widely validated [42–44].

### 3.1.3. WVC Assimilation Product

ERA5 data is fifth-generation atmospheric reanalysis data for the global climate, established by the European Center for Medium-Range Weather Forecasts (ECMWF) [45]. Developed from ERA-Interim data, ERA5 data is assimilated with worldwide observations to form a complete and consistent global dataset. It provides many meteorological elements, including hourly integrated water vapor content with a spatial resolution of 0.25°. Since the release of the ERA5 reanalysis data, many researchers have tested for accuracy [16,46]. The accuracy of the DL algorithm mainly depends on the training and test data. To improve the accuracy of the collected data, we used ERA5 data as the reference calibration data. The data can only be used when the accuracy of different data products is consistent.

### 3.1.4. Ground Observations Data

Ground-observed WVC data are also important complementary datasets. In situ WVC data from 2020 was used to build a training/testing database and evaluated existing datasets and new products. These data were obtained from Suomi Net, which is a real-time national Global Positioning System (GPS) network. A series of stations has been set up in North America. There are about 800 stations in total, and each station uses GPS signals to retrieve integrated water vapor content every 30 min, with an accuracy of 1–2 mm [47]. Table 1 lists the ground-based and satellite datasets used in this study.

**Table 1.** Datasets used in this study.

| Variable | Dataset(s) | Data Source |
|---|---|---|
| Brightness temperature (BT) <br> Land surface temperature (LST) <br> Land surface emissivity (LSE) | MOD/MYD021KM <br> MOD/MYD11 L2 <br> MOD/MYD11C1 | https://ladsweb.modaps.eosdis.nasa.gov/search/ <br> (accessed: 20 November 2021) |
| | MOD/MYD05 | https://ladsweb.modaps.eosdis.nasa.gov/search/ <br> (accessed: 10 December 2021) |
| Integrated water vapor content (WVC) | ERA5 | https://cds.climate.copernicus.eu/cdsapp#!/dataset/reanalysis-era5-single-levels?tab=form <br> (accessed: 5 January 2022) |
| | Suomi Net | https://www.cosmic.ucar.edu/what-we-do/suominet-weather-precipitation-data, (accessed: 5 January 2021) |

### 3.1.5. Data Processing

The accuracy of WVC retrieval depends on the quality of training and testing datasets. The forward physical model is used mainly to obtain the representative solution of the physical method. The parameters of the model simulation are determined, and the data accuracy is very high, but it cannot represent all the real situations, so it is necessary to supplement the solution of the statistical method. The solution of the statistical method comes from multi-source data, so to ensure the consistency of the parameters of the statistical solution it is necessary to ensure that the collected data are synchronized in time and space. To ensure the quality of collected MODIS data products, we have eliminated invalid data due to cloud coverage and other reasons according to the quality control documents of MODIS (including MOD/MYD05 and MOD/MYD11). With ERA5 data as a reference, all data were collected at a resolution of 0.25°, and the time was consistent. The data will be selected only when the accuracy of the two integrated water vapor content products is consistent.

### 3.2. Methods

In order to make the algorithm for retrieving WVC based on DL have physical significance and form a general paradigm, here we derive the physical and statistical methods based on the radiative transfer energy balance equation, and elaborate the coupling mechanism of the formation of the paradigm.

### 3.2.1. The Physical Method

The long wave heat radiation transfer process is shown in Figure 2. The derivation of the physical method for WVC retrieval is based on the thermal radiance of the ground and its transfer from the ground through the atmosphere to the remote sensor. Generally speaking, the ground is not a blackbody, and ground emissivity has to be considered for computing the thermal radiance emitted by the ground. The atmosphere has important effects on the received radiance at remote sensor level.

The thermal infrared band is mainly affected by the WVC, which attenuates the thermal radiation energy emitted from the ground. Under clear sky conditions, surface energy radiation changes with different surface types. LST affects the change in near-surface temperature. The absorption of heat by integrated water vapor content will change the atmospheric profile temperature, which will affect the integrated water vapor content saturation. In addition, due to the continuous exchange of energy in the earth–atmosphere system, the vertical density distribution of the atmosphere is also uneven. The upward and downward transmissions are different. Therefore, atmospheric radiation is often expressed in two forms (upward and downward). The radiance value of the sensor at a certain angle can be expressed as Equation (1).

$$B_i(T_i) = \varepsilon_i B_i(T_s) \tau_i(WVC, \theta) + R_i^{\uparrow}(\theta) + (1 - \varepsilon_i) \tau_i(WVC, \theta) R_i^{\downarrow}(\theta) \tag{1}$$

where $T_i$ is the BT at the satellite of the band $i$, $B_i(T_i)$ is the radiation received by the sensor, $B_i(T_s)$ represents the ground radiation, $T_s$ is the LST, $\tau_i(WVC, \theta)$ is the atmospheric transmittance in band $i$ at the observation angle θ, $\varepsilon_i$ is the LSE in the band $i$, and $R_i^\downarrow(\theta)$ and $R_i^\uparrow(\theta)$ are the downward and upward atmospheric radiances, respectively.

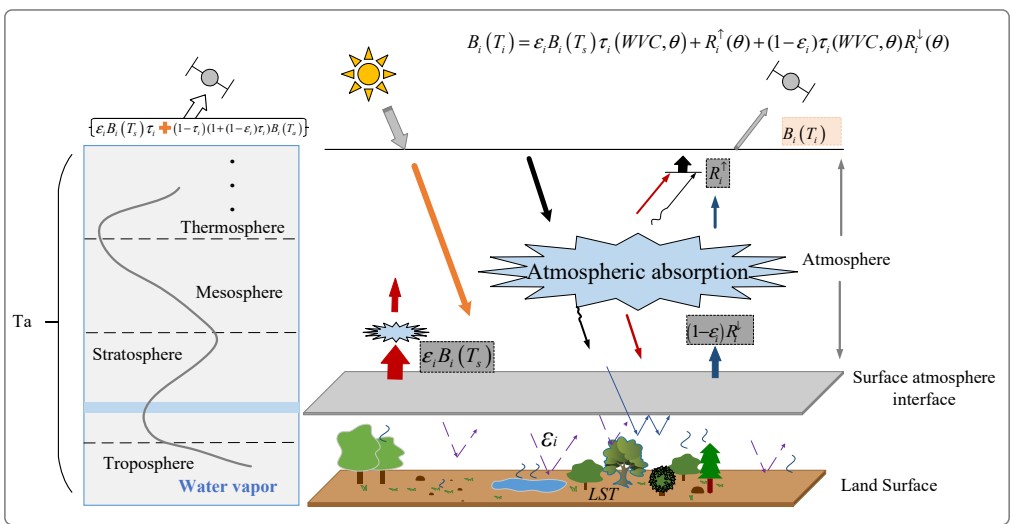

**Figure 2.** Simplified diagram of radiative transfer modeling relating LST, LSE and WVC.

In Franc and Cracknell [14], the upward atmospheric radiation $R_i^\uparrow(\theta)$ and downward atmospheric radiation $R_i^\downarrow(\theta)$ were expressed as:

$$R_i^\uparrow(\theta) = \int_0^Z B_i(T_z) \frac{\partial \tau_i(\theta, z, Z)}{\partial z} dz \qquad (2)$$

$$R_i^\downarrow(\theta) = 2 \int_0^{\frac{\pi}{2}} \int_h^0 B_i(T_z) \frac{\partial \tau_i'(\theta', z, 0)}{\partial z} \cos\theta' \sin\theta' dz d\theta' \qquad (3)$$

where $T_z$ is the atmospheric temperature at height z, Z is the sensor height, $\theta$ is the upward direction of atmospheric radiation, and $\tau_i(\theta, z, Z)$ represents the upward atmospheric transmittance from height z to the sensor. $\theta'$ is the downward direction of atmospheric radiation, $h$ is the atmospheric height, and $\tau'_i(\theta', z, 0)$ represents the downward atmospheric transmittance from height z to the surface. Qin et al. [48] used the median value theorem to process the upward and downward atmospheric radiation. With this we obtain

$$R_i^\uparrow(\theta) = (1 - \tau_i(WVC, \theta))B_i(T_a) \qquad (4)$$

$$R_i^\downarrow(\theta) = (1 - \tau_i(WVC, \theta))B_i(T_a^\downarrow) \qquad (5)$$

where $T_a$ and $T_a^\downarrow$ are the effective average atmospheric temperature upward and downward respectively, $B_i(T_a)$ and $B_i\left(T_a^\downarrow\right)$ represents the effective average atmospheric radiation of $T_a$ and $T_a^\downarrow$ in band $i$. Next, Qin [48] found that substituting $T_a$ for $T_a^\downarrow$ had no significant impact on accuracy. Therefore, Equation (1) can be written as,

$$B_i(T_i) = \varepsilon_i B_i(T_s)\tau_i(WVC, \theta) + (1 + (1 - \varepsilon_i)\tau_i(WVC, \theta))(1 - \tau_i(WVC, \theta))B_i(T_a) \qquad (6)$$

Equation (6) indicates that the radiative transfer formula of each band has four unknowns ($\tau_i$, $T_s$, $T_a$, $\varepsilon_i$). For N bands, there will be 2N + 2 unknowns (τ and ε of N bands, $T_s$ and $T_a$). We need to reduce variables according to the geophysical relationship between them. The atmospheric transmittance of the TIR band is mainly affected by the

integrated water vapor content and other gases (*O*), and the transmittance can be expressed as Equation (7).

$$\tau_i(WVC, \theta) = f(WVC, O) \tag{7}$$

The change in thermal infrared band transmittance is mainly affected by the change in integrated water vapor content, because other gas components in the atmosphere are relatively stable. Therefore, the transmittance of the TIR band is mainly a function of integrated water vapor content change. The emissivity of different bands is different, but the emissivity curve of surface type is stable. Therefore, as long as the ground surface type (G) is determined, the emissivity change curve can be determined. Therefore, the emissivity of different bands is a function of the surface type, which can be expressed by Equation (8).

$$\varepsilon_i = f(G) \tag{8}$$

From the above derivation and analysis, Equation (6) can be simplified into four unknowns (WVC, LST, G, and $T_a$). To solve the equations, at least four TIR bands are needed to construct the equations. Mao, et al. [49] found that there was an approximate restrictive relationship between the average atmospheric action temperature ($T_a$) and the brightness temperature ($T_i$) at satellite, as shown in Equation (9).

$$T_a \approx A + B \times T_i \tag{9}$$

where A and B are constants that are limited by geographical conditions. However, the constraint relation cannot be determined exactly, which leaves uncertainty in the solution by traditional methods. If the calculation method is optimized by deep learning, the equation can be reduced to three unknowns (WVC, LST and G), which means that the equation set can be constructed with only three bands at least, but the accuracy may be slightly affected.

In summary, the number of unknowns in Equation (6) has been reduced to four (WVC, LST, G, and $T_a$). In mathematical equations, if the only solution is found, the number of equations is greater than or equal to n (here there are four unknowns). Therefore, when using physical methods to retrieve WVC, at least four TIR bands are needed to construct the equations. Since high-precision LST and LSE products can be retrieved, we used LST and LSE as prior knowledge to improve the stability and accuracy of the WVC inversion algorithm. In addition, in the previous derivation, we found that, if we use the relationship between the average effective temperature of the atmosphere and the BT at satellite which ensures that the BT information on the satellite mainly comes from the atmosphere, we can also use only three TIR bands to construct the retrieval equation. The above analysis was the process that we used, utilizing physical logic reasoning to determine the minimum number of input parameters required by DL.

### 3.2.2. The Statistical Method

Theoretically, if we can find all the representative solutions of physical methods in the real world, we can directly use the representative solutions of physical methods as the training and test database for DL to perform retrieval optimization calculations. In fact, the forward model MODTRAN cannot simulate all situations, especially the mixed pixel situation. Therefore, we need to collect other representative solutions, which we call statistical solutions. This statistical method also needs to meet the basic conditions of physical methods, that is, at least four or three bands of BT as input parameters and of WVC as output parameters, and also needs to obtain the corresponding LST and LLSE information. Therefore, the fuzzy function of the statistical method can be expressed as Equation (10).

$$WVC = \sum_{i=1}^{n} \Pi f(T_i) + b \tag{10}$$

where $f(T_i)$ is a polynomial function of BT, $n$ ($n \geq 3$) is the number of TIR bands used for retrieval, $\Pi$ is an empirical coefficient, which can be determined by the contribution of the

infrared band in WVC retrieval, and *b* is a constant. Statistical methods are essentially the same as physical methods, and the solutions with statistical methods are obtained from multi-source data. The statistical solution is a beneficial supplement to the physical method solution, especially in the case of mixed pixels.

### 3.2.3. Physics-Statistical Coupling Deep Learning Based on Prior Knowledge

The logical derivation of physical methods is to determine that there is a unique mathematical function relationship between the input parameters and the output parameters of DL. The statistical method is to supplement the lack of representativeness of the simulation solution of the physical forward model. We combined the solutions of physical methods and statistical methods into the training and testing database of DL, thus realizing the purpose of coupling physical and statistical methods. The DNN can model complex nonlinear relationships between input and output variables [50]. The DNN structure is shown in Figure 3; a complete DNN includes an input layer, an output layer, and a number of hidden layers. The activation function can not only introduce nonlinear expression ability, but also improve the robustness of the model and alleviate the problem of gradient disappearance. In this study, we chose the common function 'Sigmoid' as the activation function of DNN (shown in Equation (11), where *x* is the output value of the previous hidden layer node). The representation of the function in the neural network is carried out by a group of independent neurons, which learn from the input during the training phase. After all patterns had been presented, the interconnecting synaptic weight of each neuron was adjusted [51]. The DNN constructed in this study adopted Kalman filter technology, which greatly improves the convergence speed in the learning stage and the separability of nonlinear boundary problems. Moreover, the training time was greatly shortened and the precision of the neural network was improved [52].

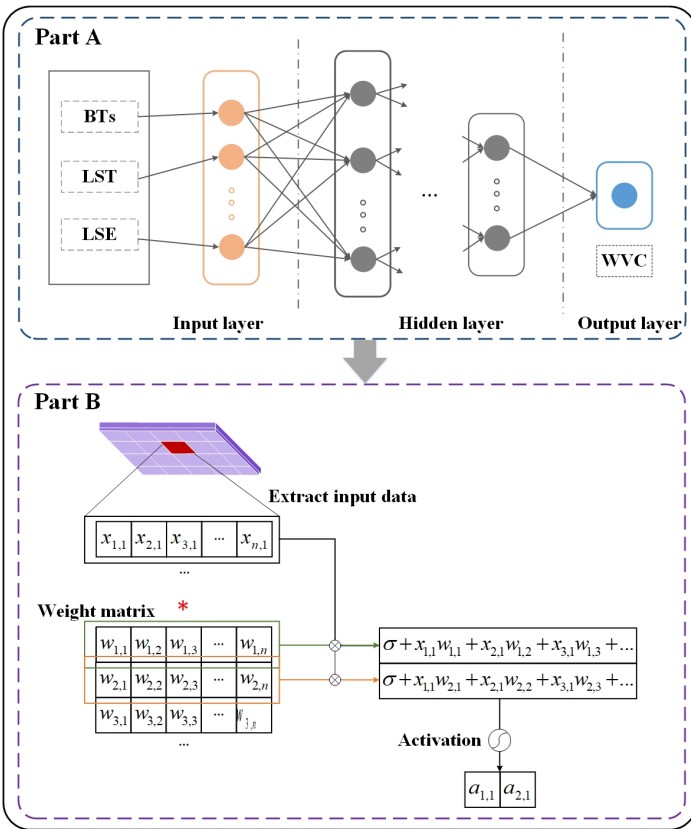

**Figure 3.** DNN and schematic of neural network operation. (*n* is the number of input parameters, $x_{i,j}$ are the input parameters of the neural network, and WVC is the output parameter. $w_{i,j}$ is the weight of each neuron and $\sigma$ is the bias. $a_{i,j}$ is the parameter input to the next layer of neurons after the activation function of the previous layer of neurons, $*$ indicates multiplication).

$$f(x) = \frac{1}{1 + e^{-x}} \tag{11}$$

### 3.2.4. Physics-Statistical Coupled Deep Learning Retrieval Scheme

A specific scheme for retrieving WVC using the PSP–DL model was presented. Firstly, we used MODTRAN simulated atmospheric parameters (training data: 126,720, test data: 59,400) to verify the availability of the physical method, and obtained the theoretical accuracy of WVC retrieval and the optimal scheme for MODIS retrieval of atmospheric water vapor. Secondly, the accuracy of the statistical method was verified by adding reliable WVC data (training data: 25,805, test data: 11,803). Radiometric calibration, geometric correction and atmospheric correction were used to preprocess MODIS images to obtain more accurate spectral information, and to obtain the brightness temperature of MODIS and the WVC of the corresponding image.

According to the band design purpose of MODIS and the position of the band center wavelength, we have determined a representative band combination. Two combination models were constructed (Table 2). The first was a combination with prior knowledge (BT in thermal infrared bands 27, 28, 29, 31 and 32, and LST and LSE in bands 29, 31 and 32). The second was the combination without prior knowledge (BT in thermal infrared bands 27, 28, 29, 31 and 32).

**Table 2.** Band combinations used for retrieval of WVC.

| Model | Situation | Variables |
|:---:|:---:|:---:|
| 1 | Use prior knowledge | BTs in thermal infrared band 27/28/29/31/32, LST and LSE of bands 29, 31, 32 |
| 2 | No prior knowledge used | BTs in thermal infrared band 27/28/29/31/32 |

## 4. Results and Validation

### 4.1. Theoretical Accuracy Validation and Analysis

We demonstrated the solution (retrieval) of the physical method (Equation (6)) using simulation data and DL. The information obtained by satellite sensors includes surface radiation and reflection information, which is mainly determined by the type of surface. Therefore, we also used the prior knowledge of LST and LSE to solve Equation (6) of WVC, i.e., used accurate and reliable prior knowledge with high-precision BTs as the input from DL to retrieve WVC. We have carried out inversion and comparative analysis on the combination with and without prior knowledge and selected the mean absolute error (MAE), root mean square error (RMSE) and $R^2$, which can represent the inversion accuracy, to find the best WVC inversion combination. The DL retrieval result with prior knowledge combination is shown in Tables 3–5 (Model 1) and the retrieval combination without prior knowledge is shown in Tables 6–8 (Model 2).

**Table 3.** WVC retrieval results using LST and LSE and bands 27, 29, 31, and 32.

| LST and LSE + 4BTs (from 27, 29, 31, 32) → WVC | | | | | | | | | | | | | | |
|:---:|:---:|:---:|:---:|:---:|:---:|:---:|:---:|:---:|:---:|:---:|:---:|:---:|:---:|:---:|
| Nodes | | 400 | | | 500 | | | 600 | | | 700 | | | 800 | |
| Layers | MAE | RMSE | $R^2$ | MAE | RMSE | $R^2$ | MAE | RMSE | $R^2$ | MAE | RMSE | $R^2$ | MAE | RMSE | $R^2$ |
| 3 | 0.11 | 0.15 | 0.980 | 0.18 | 0.24 | 0.949 | 0.11 | 0.15 | 0.979 | 0.11 | 0.14 | 0.981 | 0.10 | 0.13 | 0.985 |
| 4 | 0.10 | 0.12 | 0.986 | 0.12 | 0.15 | 0.978 | 0.09 | 0.12 | 0.987 | 0.08 | 0.11 | 0.989 | 0.09 | 0.11 | 0.988 |
| 5 | 0.11 | 0.15 | 0.980 | 0.11 | 0.15 | 0.979 | 0.11 | 0.14 | 0.982 | 0.08 | 0.10 | 0.990 | 0.11 | 0.14 | 0.981 |
| 6 | 0.12 | 0.16 | 0.977 | 0.12 | 0.16 | 0.977 | 0.10 | 0.12 | 0.986 | 0.08 | 0.10 | 0.990 | 0.08 | 0.10 | 0.991 |
| 7 | 0.14 | 0.2 | 0.964 | 0.12 | 0.16 | 0.977 | 0.12 | 0.16 | 0.978 | 0.11 | 0.14 | 0.981 | 0.09 | 0.12 | 0.988 |
| 8 | 0.09 | 0.12 | 0.987 | 0.10 | 0.13 | 0.985 | 0.11 | 0.14 | 0.982 | 0.09 | 0.12 | 0.988 | 0.10 | 0.13 | 0.985 |
| 9 | 0.12 | 0.16 | 0.975 | 0.13 | 0.18 | 0.970 | 0.09 | 0.12 | 0.987 | 0.09 | 0.12 | 0.986 | 0.09 | 0.13 | 0.985 |
| 10 | 0.17 | 0.26 | 0.940 | 0.10 | 0.13 | 0.985 | 0.10 | 0.14 | 0.983 | 0.08 | 0.11 | 0.989 | **0.07** | **0.10** | **0.991** |

**Table 4.** WVC retrieval results using LST and LSE and bands 28, 29, 31, and 32.

| | LST and LSE + 4BTs (from 28, 29, 31, 32) → WVC | | | | | | | | | | | | | | |
|---|---|---|---|---|---|---|---|---|---|---|---|---|---|---|---|
| **Nodes** | **400** | | | **500** | | | **600** | | | **700** | | | **800** | | |
| **Layers** | **MAE** | **RMSE** | **R²** | **MAE** | **RMSE** | **R²** | **MAE** | **RMSE** | **R²** | **MAE** | **RMSE** | **R²** | **MAE** | **RMSE** | **R²** |
| 3 | 0.12 | 0.15 | 0.978 | 0.15 | 0.20 | 0.964 | 0.11 | 0.15 | 0.979 | 0.12 | 0.16 | 0.977 | 0.12 | 0.15 | 0.978 |
| 4 | 0.13 | 0.17 | 0.974 | 0.13 | 0.17 | 0.974 | 0.14 | 0.19 | 0.969 | 0.10 | 0.13 | 0.985 | 0.12 | 0.15 | 0.979 |
| 5 | 0.13 | 0.17 | 0.972 | 0.13 | 0.17 | 0.972 | 0.14 | 0.19 | 0.963 | 0.14 | 0.19 | 0.963 | 0.11 | 0.15 | 0.978 |
| 6 | 0.14 | 0.18 | 0.966 | 0.11 | 0.14 | 0.980 | **0.09** | **0.12** | **0.987** | 0.11 | 0.14 | 0.981 | 0.11 | 0.14 | 0.981 |
| 7 | 0.24 | 0.32 | 0.906 | 0.13 | 0.17 | 0.974 | 0.13 | 0.17 | 0.974 | 0.14 | 0.18 | 0.967 | 0.11 | 0.14 | 0.981 |
| 8 | 0.13 | 0.18 | 0.970 | 0.11 | 0.14 | 0.978 | 0.13 | 0.18 | 0.972 | 0.13 | 0.16 | 0.973 | 0.11 | 0.15 | 0.978 |
| 9 | 0.17 | 0.23 | 0.951 | 0.15 | 0.20 | 0.962 | 0.11 | 0.14 | 0.982 | 0.11 | 0.14 | 0.980 | 0.09 | 0.12 | 0.984 |
| 10 | 0.11 | 0.14 | 0.980 | 0.11 | 0.14 | 0.980 | 0.12 | 0.16 | 0.978 | 0.10 | 0.13 | 0.983 | 0.10 | 0.13 | 0.983 |

**Table 5.** WVC retrieval results using LST and LSE and bands 27, 28, 31, and 32.

| | LST and LSE + 4BTs (from 27, 28, 31, 32) → WVC | | | | | | | | | | | | | | |
|---|---|---|---|---|---|---|---|---|---|---|---|---|---|---|---|
| **Nodes** | **400** | | | **500** | | | **600** | | | **700** | | | **800** | | |
| **Layers** | **MAE** | **RMSE** | **R²** | **MAE** | **RMSE** | **R²** | **MAE** | **RMSE** | **R²** | **MAE** | **RMSE** | **R²** | **MAE** | **RMSE** | **R²** |
| 3 | 0.17 | 0.25 | 0.944 | 0.12 | 0.17 | 0.972 | 0.09 | 0.12 | 0.986 | 0.09 | 0.13 | 0.985 | 0.07 | 0.10 | 0.990 |
| 4 | 0.11 | 0.16 | 0.960 | 0.14 | 0.23 | 0.969 | 0.08 | 0.12 | 0.986 | 0.08 | 0.11 | 0.982 | 0.06 | 0.09 | 0.989 |
| 5 | 0.07 | 0.10 | 0.979 | 0.16 | 0.30 | 0.938 | 0.09 | 0.14 | 0.985 | 0.07 | 0.12 | 0.984 | 0.08 | 0.12 | 0.986 |
| 6 | 0.08 | 0.11 | 0.987 | 0.12 | 0.20 | 0.952 | 0.08 | 0.11 | 0.986 | 0.07 | 0.20 | 0.975 | 0.06 | 0.08 | 0.988 |
| 7 | 0.10 | 0.22 | 0.955 | 0.11 | 0.31 | 0.952 | 0.07 | 0.10 | 0.989 | 0.09 | 0.16 | 0.970 | **0.05** | **0.07** | **0.993** |
| 8 | 0.13 | 0.32 | 0.940 | 0.10 | 0.40 | 0.908 | 0.09 | 0.12 | 0.987 | 0.07 | 0.10 | 0.981 | 0.06 | 0.11 | 0.989 |
| 9 | 0.11 | 0.16 | 0.904 | 0.12 | 0.18 | 0.939 | 0.06 | 0.09 | 0.988 | 0.05 | 0.07 | 0.990 | 0.05 | 0.08 | 0.990 |
| 10 | 0.12 | 0.18 | 0.966 | 0.07 | 0.10 | 0.975 | 0.05 | 0.11 | 0.986 | 0.06 | 0.09 | 0.992 | 0.05 | 0.07 | 0.994 |

**Table 6.** WVC retrieval results using bands 27, 29, 31, and 32.

| | 4BTs (from 27, 29, 31, and 32) → WVC | | | | | | | | | | | | | | |
|---|---|---|---|---|---|---|---|---|---|---|---|---|---|---|---|
| **Nodes** | **400** | | | **500** | | | **600** | | | **700** | | | **800** | | |
| **Layers** | **MAE** | **RMSE** | **R²** | **MAE** | **RMSE** | **R²** | **MAE** | **RMSE** | **R²** | **MAE** | **RMSE** | **R²** | **MAE** | **RMSE** | **R²** |
| 3 | 0.19 | 0.25 | 0.943 | 0.18 | 0.24 | 0.948 | 0.17 | 0.23 | 0.952 | 0.17 | 0.23 | 0.952 | 0.17 | 0.23 | 0.950 |
| 4 | 0.19 | 0.25 | 0.941 | 0.17 | 0.22 | 0.953 | 0.17 | 0.23 | 0.950 | 0.17 | 0.23 | 0.95 | 0.17 | 0.23 | 0.952 |
| 5 | 0.17 | 0.23 | 0.950 | 0.20 | 0.29 | 0.921 | 0.20 | 0.28 | 0.926 | 0.18 | 0.25 | 0.942 | 0.17 | 0.22 | 0.953 |
| 6 | 0.17 | 0.23 | 0.951 | 0.17 | 0.23 | 0.950 | 0.16 | 0.22 | 0.953 | 0.18 | 0.25 | 0.941 | 0.17 | 0.23 | 0.950 |
| 7 | 0.17 | 0.24 | 0.946 | 0.17 | 0.23 | 0.948 | 0.19 | 0.27 | 0.934 | 0.18 | 0.24 | 0.947 | 0.18 | 0.24 | 0.945 |
| 8 | 0.19 | 0.25 | 0.942 | 0.18 | 0.23 | 0.949 | 0.18 | 0.25 | 0.939 | 0.17 | 0.23 | 0.950 | 0.17 | 0.23 | 0.952 |
| 9 | 0.19 | 0.25 | 0.941 | **0.16** | **0.22** | **0.954** | 0.17 | 0.22 | 0.953 | 0.17 | 0.24 | 0.947 | 0.17 | 0.24 | 0.944 |
| 10 | 0.19 | 0.26 | 0.936 | 0.17 | 0.23 | 0.949 | 0.18 | 0.23 | 0.949 | 0.18 | 0.24 | 0.945 | 0.18 | 0.25 | 0.944 |

**Table 7.** WVC retrieval results using bands 28, 29, 31, and 32.

| | 4BTs (from 28, 29, 31, 32) → WVC | | | | | | | | | | | | | | |
|---|---|---|---|---|---|---|---|---|---|---|---|---|---|---|---|
| **Nodes** | **400** | | | **500** | | | **600** | | | **700** | | | **800** | | |
| **Layers** | **MAE** | **RMSE** | **R²** | **MAE** | **RMSE** | **R²** | **MAE** | **RMSE** | **R²** | **MAE** | **RMSE** | **R²** | **MAE** | **RMSE** | **R²** |
| 3 | 0.22 | 0.30 | 0.918 | 0.21 | 0.29 | 0.924 | 0.20 | 0.28 | 0.927 | 0.21 | 0.28 | 0.925 | 0.21 | 0.28 | 0.927 |
| 4 | 0.21 | 0.28 | 0.926 | 0.22 | 0.29 | 0.921 | 0.21 | 0.28 | 0.926 | 0.20 | 0.28 | 0.929 | 0.20 | 0.28 | 0.927 |
| 5 | 0.20 | 0.28 | 0.928 | 0.22 | 0.29 | 0.921 | 0.20 | 0.27 | 0.930 | 0.20 | 0.28 | 0.929 | 0.21 | 0.28 | 0.925 |
| 6 | 0.21 | 0.28 | 0.926 | 0.20 | 0.28 | 0.928 | 0.20 | 0.28 | 0.927 | 0.20 | 0.28 | 0.929 | 0.21 | 0.30 | 0.917 |
| 7 | **0.20** | **0.27** | **0.931** | 0.21 | 0.29 | 0.924 | 0.20 | 0.28 | 0.928 | 0.20 | 0.27 | 0.931 | 0.21 | 0.28 | 0.926 |
| 8 | 0.25 | 0.33 | 0.896 | 0.20 | 0.28 | 0.927 | 0.21 | 0.29 | 0.923 | 0.21 | 0.28 | 0.926 | 0.21 | 0.28 | 0.926 |
| 9 | 0.22 | 0.29 | 0.920 | 0.20 | 0.28 | 0.928 | 0.23 | 0.32 | 0.907 | 0.21 | 0.28 | 0.925 | 0.21 | 0.28 | 0.925 |
| 10 | 0.21 | 0.28 | 0.926 | 0.21 | 0.28 | 0.928 | 0.21 | 0.28 | 0.928 | 0.22 | 0.29 | 0.921 | 0.20 | 0.27 | 0.930 |

**Table 8.** WVC retrieval results using bands 27, 28, 31, and 32.

| | 4BTs (from 27, 28, 31, and 32) → WVC | | | | | | | | | | | | | | |
|---|---|---|---|---|---|---|---|---|---|---|---|---|---|---|---|
| **Nodes** | **400** | | | **500** | | | **600** | | | **700** | | | **800** | | |
| **Layers** | **MAE** | **RMSE** | **R²** | **MAE** | **RMSE** | **R²** | **MAE** | **RMSE** | **R²** | **MAE** | **RMSE** | **R²** | **MAE** | **RMSE** | **R²** |
| 3 | 0.12 | 0.17 | 0.974 | 0.08 | 0.14 | 0.981 | 0.08 | 0.11 | 0.989 | 0.06 | 0.08 | 0.994 | 0.06 | 0.13 | 0.985 |
| 4 | 0.09 | 0.14 | 0.979 | 0.08 | 0.21 | 0.974 | 0.09 | 0.34 | 0.966 | 0.07 | 0.18 | 0.973 | 0.05 | 0.22 | 0.980 |
| 5 | 0.14 | 0.31 | 0.952 | 0.05 | 0.07 | 0.962 | 0.06 | 0.17 | 0.940 | 0.07 | 0.14 | 0.979 | 0.06 | 0.14 | 0.974 |
| 6 | 0.06 | 0.09 | 0.918 | **0.06** | **0.08** | **0.997** | 0.08 | 0.41 | 0.915 | 0.05 | 0.10 | 0.986 | 0.07 | 0.22 | 0.976 |
| 7 | 0.10 | 0.53 | 0.821 | 0.07 | 0.21 | 0.967 | 0.06 | 0.15 | 0.953 | 0.05 | 0.10 | 0.989 | 0.18 | 1.33 | 0.450 |
| 8 | 0.07 | 0.18 | 0.888 | 0.18 | 1.17 | 0.413 | 0.08 | 0.31 | 0.946 | 0.06 | 0.16 | 0.984 | 0.11 | 0.60 | 0.548 |
| 9 | 0.09 | 0.24 | 0.974 | 0.08 | 0.22 | 0.420 | 0.06 | 0.15 | 0.951 | 0.18 | 1.07 | 0.472 | 0.07 | 0.15 | 0.814 |
| 10 | 0.05 | 0.07 | 0.947 | 0.10 | 0.28 | 0.936 | 0.07 | 0.18 | 0.986 | 0.05 | 0.07 | 0.513 | 0.05 | 0.18 | 0.978 |

For model 1, WVC retrieval accuracy was good and stable using four and five bands (results shown in Tables 3–5). By comparing these results, it was found that the retrieval accuracy using band 27 was better than that using band 28. This is likely to be due to the strong absorption zone of integrated water vapor content in band 27, which can include more water vapor information for the whole layer. The highest accuracy was obtained with the 27, 28, 31, and 32 band combination, which yielded an MAE = 0.05 g/cm$^2$, RMSE = 0.07 g/cm$^2$, and R$^2$ = 0.993, when the number of hidden layers was seven and the number of nodes per layer was 800. Through analysis, WVC retrieval accuracy of the band combination 27, 28, 29, 31, and 32 was similar to retrieval with band 27, 28, 31 and 32 with the use of prior knowledge. Therefore, data redundancy and retrieval band quality should also be taken into account.

Model 2 was designed to compare the effect of using prior knowledge on the retrieval accuracy, and used the same band combinations as model 1, but without prior knowledge input, and the retrieval process was the same as for model 1. In order to find the retrieval results with the highest accuracy, different combinations of hidden layers and number of neurons were tried (results shown in Tables 6–8). Compared with Tables 3 and 6, the minimum MAE of WVC retrieval using LST and LSE and BTs of bands 27, 29, 31 and 32 as input parameters was 0.07 g/cm$^2$, and the minimum MAE of retrieval error using only BTs of bands 27, 29, 31 and 32 as input parameters was 0.16 g/cm$^2$, which shows that the increase of prior knowledge reduces MAE by 0.09 g/cm$^2$. Compared with Tables 4 and 7, when bands 28, 29, 31 and 32 were combined, the MAE with prior knowledge was 0.11 g/cm$^2$ lower than that without prior knowledge. It can be seen from Table 8 that, when the number of hidden layers was 6 and the number of nodes in each layer was 500, the minimum retrieval error was the combination of bands 27, 28, 31 and 32, MAE = 0.06 g/cm$^2$, RMSE = 0.08 g/cm$^2$, R$^2$ = 0.997. From the comparison between Tables 5 and 8, it is indicated that, for the combination with 27, 28, 31 and 32, the error with prior knowledge was only 0.01 g/cm$^2$ lower than that without prior knowledge. The main reason is that the BT of bands 27 and 28 is very sensitive to the integrated water vapor content and is little affected by the ground. Therefore, for the TIR band whose band combination of BT at satellite is greatly affected by the ground, the retrieval accuracy can be greatly improved by using prior knowledge. On the contrary, for the TIR band combination that is less affected by the ground and mainly affected by the integrated water vapor content, the ground prior knowledge may not be required.

The accuracy of WVC retrieved by DL for different combinations was generally higher than that of traditional methods. To better evaluate the error distribution of different model combinations, we have analyzed the error distribution, as shown in Figures 4 and 5. The error of model 1 was relatively small, the error fluctuation range was small, and the retrieval result was stable. Due to the high accuracy of retrieval using prior knowledge, data points basically overlap. Taking 2 g/cm$^2$ as the center, the retrieval error increases at both ends, mainly because the amount of training and test data at both ends was relatively small. From the analysis, we can see that the retrieval error using prior knowledge was more stable and smaller. When there are high-precision LST and LSE products, prior knowledge is a better choice. However, if there is no LST and LSE or their accuracy cannot be guaranteed, prior knowledge can also be omitted. In addition, although the accuracy of the retrieval combination of five thermal bands was slightly higher than that of four bands, considering the instrument band setting, we recommend using four bands. In practical application, we should consider the availability, accuracy and redundancy of data according to the thermal infrared band setting of the sensor. In the future, the instrument band design can be optimized according to our analysis.

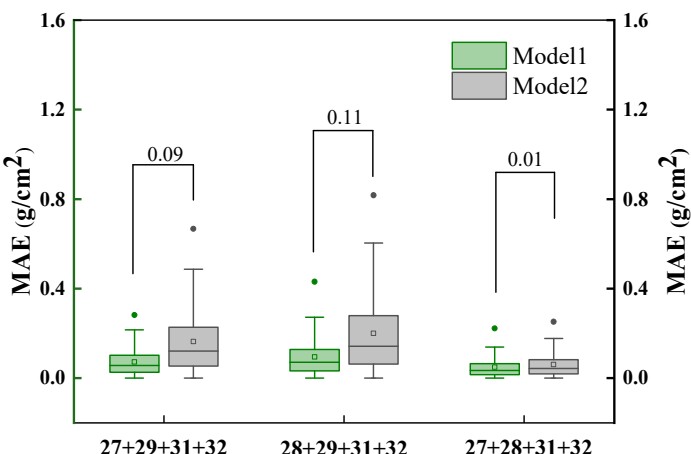

**Figure 4.** Boxplot of absolute error of retrieval result. The left side of the connecting line shows the errors from model 1, while the right side shows the errors from model 2.

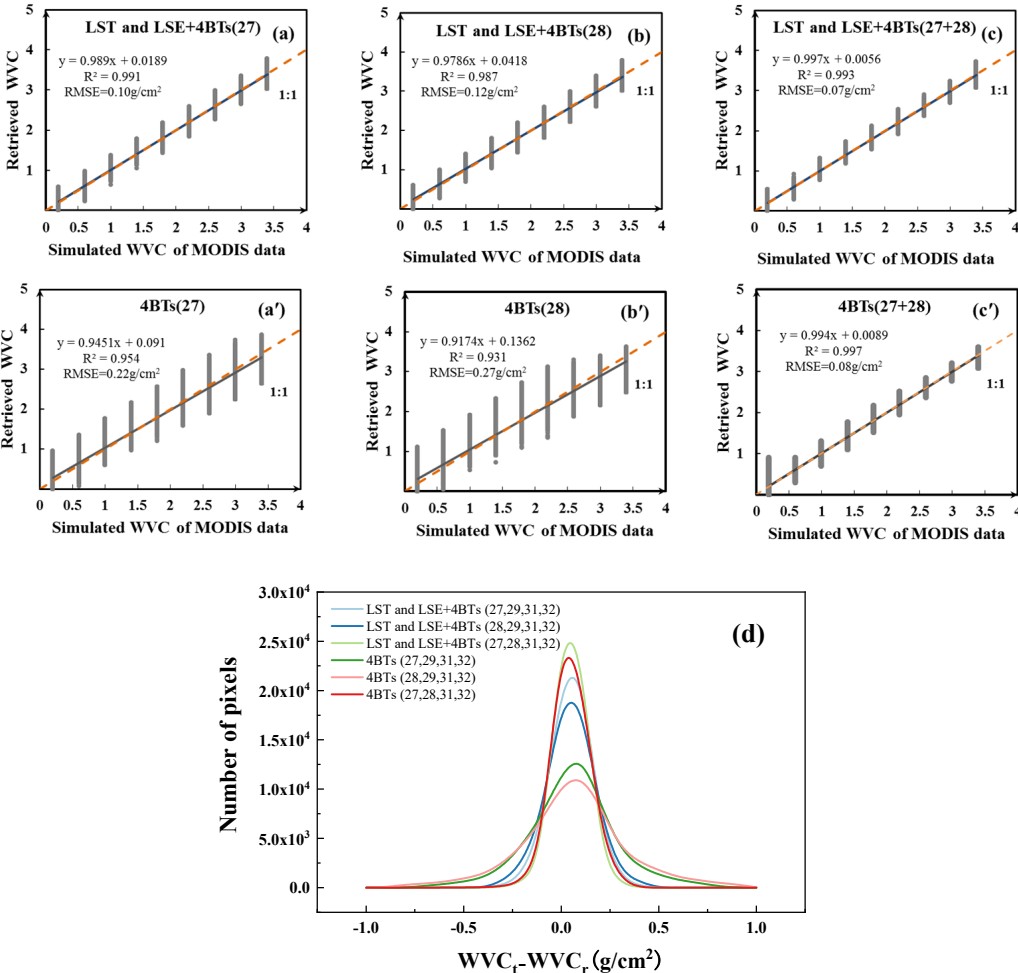

**Figure 5.** Retrieval results of PSP–DL. WVC$_r$ shows retrieval results, WVC$_t$ is the MODTRAN simulation data. (**a**–**c**) are scatter plots of model 1 retrieval results and simulation data, (**a**) is the combination of prior knowledge and bands 27, 29, 31 and 32, (**b**) is the combination of prior knowledge and bands 28, 29, 31 and 32, (**c**) is the combination of prior knowledge and bands 27, 28, 31 and 32. (**a′**–**c′**) are scatter plots of model 1 retrieval results and simulation data, (**a′**) is the combination of bands 27, 29, 31 and 32, (**b′**) is the combination of bands 28, 29, 31 and 32, (**c′**) is the combination of bands 27, 28, 31 and 32, (**d**) is the error distribution of simulated data and retrieval data.

*4.2. Practical Validation and Analysis*

The simulation data mainly represent pure pixels, so to make the retrieval results more representative, we supplemented with corresponding high-precision statistical data. The retrieval calculation and accuracy evaluation, after the simulation data were supplemented with statistical data, were similar to the analysis in Section 4.1 and will not be repeated here. So far, the method for retrieving integrated water vapor content using LST and LSE as prior knowledge and DL coupled physical and statistical methods has been completed. MODIS image of southern North America was selected as the case study, and the study scope is shown as Figure 6. The study area is located within 14°–49°N and 66°–124°W, and features a variety of land surface types, including deserts, basins, plains, hills, plateaus, and mountains. The terrain is relatively high in average elevation in the west and low in the east. The continental topography is characterized by north–south mountain ranges on both east and west coasts. Temperate maritime climate with abundant rainfall and Mediterranean climate are characteristic of the coastal mountains over much of the west coast. The windward slope is affected by the cold ocean current, keeping temperatures cooler, while the Mediterranean climate has less rain in the high-temperature period and more rain in the low-temperature period. The central plain of the United States has a temperate continental climate, and the precipitation changes dramatically with the seasons. The eastern and southern regions have a subtropical monsoon humid climate, and are the sites of many ground observations and the prior knowledge information used in this study.

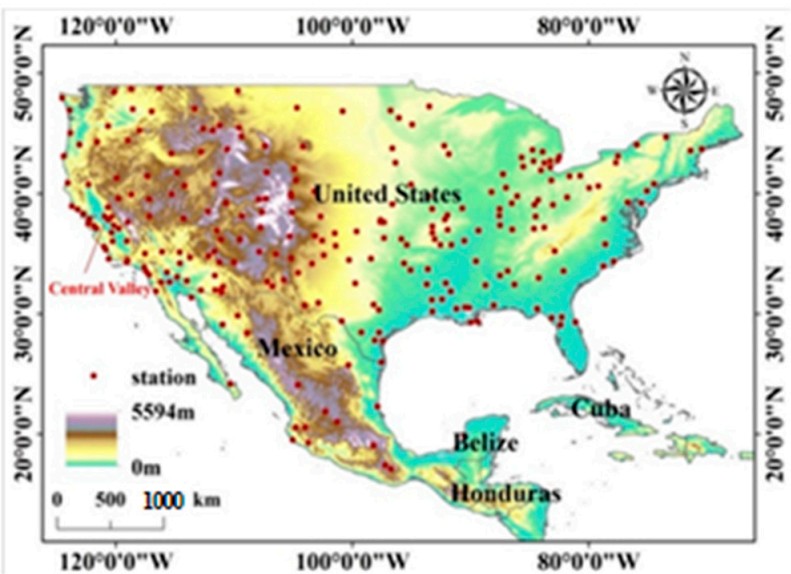

**Figure 6.** A case study of southern North America (The United States and Mexico).

4.2.1. Cross Validation

Two MODIS images were selected for the PSP–DL method, to demonstrate the retrieval of WVC. One was a MODIS VNIR integrated water vapor content product image dated 10 August 2020, and the other was a MODIS TIR integrated water vapor content image from the same day. The training database used MOD/YD 021KM image data and corresponding WVC products, and the image data ensured that there were valid data every month. The band combination with the best accuracy was selected for verification, namely the BTs of 27, 28, 31, and 32 thermal infrared bands, with LST and LSE used as input parameters of PSP–DL.

To comprehensively evaluate the accuracy of the PSP–DL method, we compared the WVC results retrieved by PSP–DL with MODIS water vapor products. Figures 7 and 8 show that the WVC results retrieved by PSP–DL were consistent with the general trends of the WVC products, and that WVC increased gradually from northwest to southeast, consistent with the distribution of climate regions. However, the accuracy of retrieval was

low in relatively dry and humid environments. The main reason here is that there is no continuous data at both ends of the function curve when using DL to solve the function curve of the solution of the equation system, so the accuracy at both ends is slightly lower.

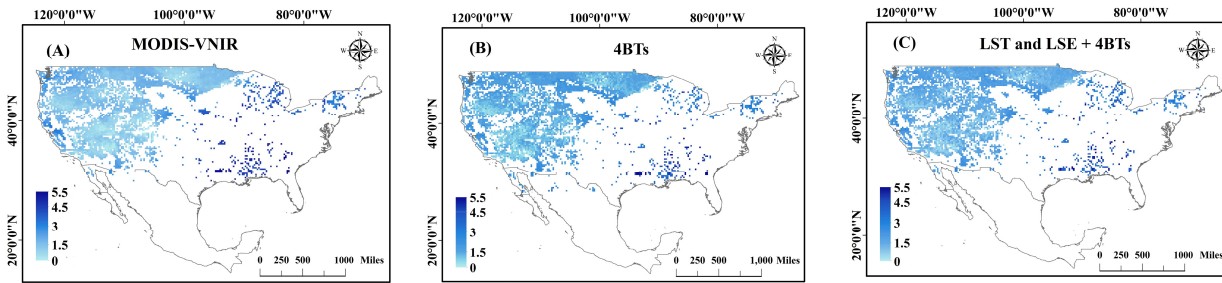

**Figure 7.** WVC (g/cm$^2$) retrieved by PSP–DL method. Integrated water vapor content products: (**A**) MODIS-VNIR, (**B**) WVC retrieved from MODIS bands 27, 28, 31 and 32, (**C**) WVC retrieved from MODIS bands 27, 28, 31 and 32 and MODIS LST and LSE products (MOD11). The white areas indicate invalid values.

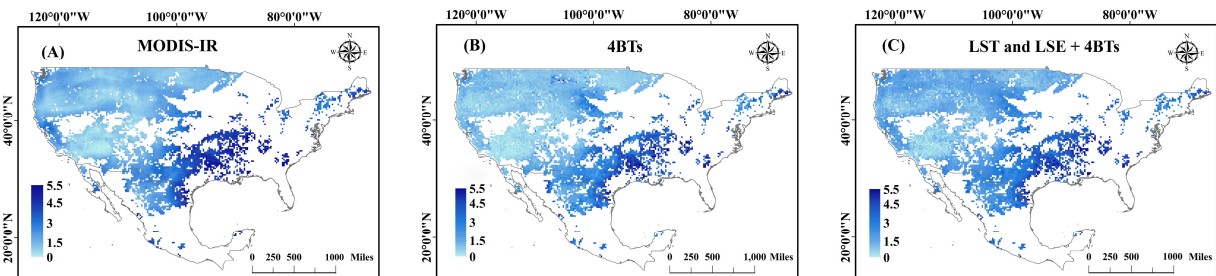

**Figure 8.** WVC (g/cm$^2$) retrieved by PSP–DL method. Integrated water vapor content products: (**A**) MODIS-TIR, (**B**) WVC retrieved from MODIS bands 27, 28, 31 and 32, (**C**) WVC retrieved from MODIS bands 27, 28, 31 and 32 and MODIS LST and LSE products (MOD11). The white areas indicate invalid values.

Figures 9 and 10 are the results of image cross-validation, which show a good overall spatial comparison between different products. Further analysis revealed that the combination of using four bands with prior knowledge was better than the combination using four bands alone. The error between the two WVC products and the WVC retrieved by the PSP–DL method was mostly between ($-0.5$ g/cm$^2$, 0.5 g/cm$^2$). Compared to the MODIS-derived WVC, the highest MAE and RMSE of WVC retrieved by DL method were 0.22 g/cm$^2$ and 0.28 g/cm$^2$. The values with poor retrieval accuracy were largely at the boundaries of cloud pixels, because some of these pixels contain cloud. The central valley of the northern coastal mountains was affected by the subtropical high-pressure belt, and the summer was characteristically hot and dry. The area was mainly irrigated agriculture, and irrigation promotes evapotranspiration, which increased atmospheric humidity during the growing season and summer. The change of LSE after irrigation can lead to a decrease in LST retrieval accuracy, which will reduce the retrieval accuracy of WVC. Comparative analysis showed that our algorithm has a high consistency with other products, and had certain advantages over those other products, such as situational adaptation through the supplementing of high-precision samples. To further improve the accuracy, we can build training and testing databases for different regions and add more high-precision data.

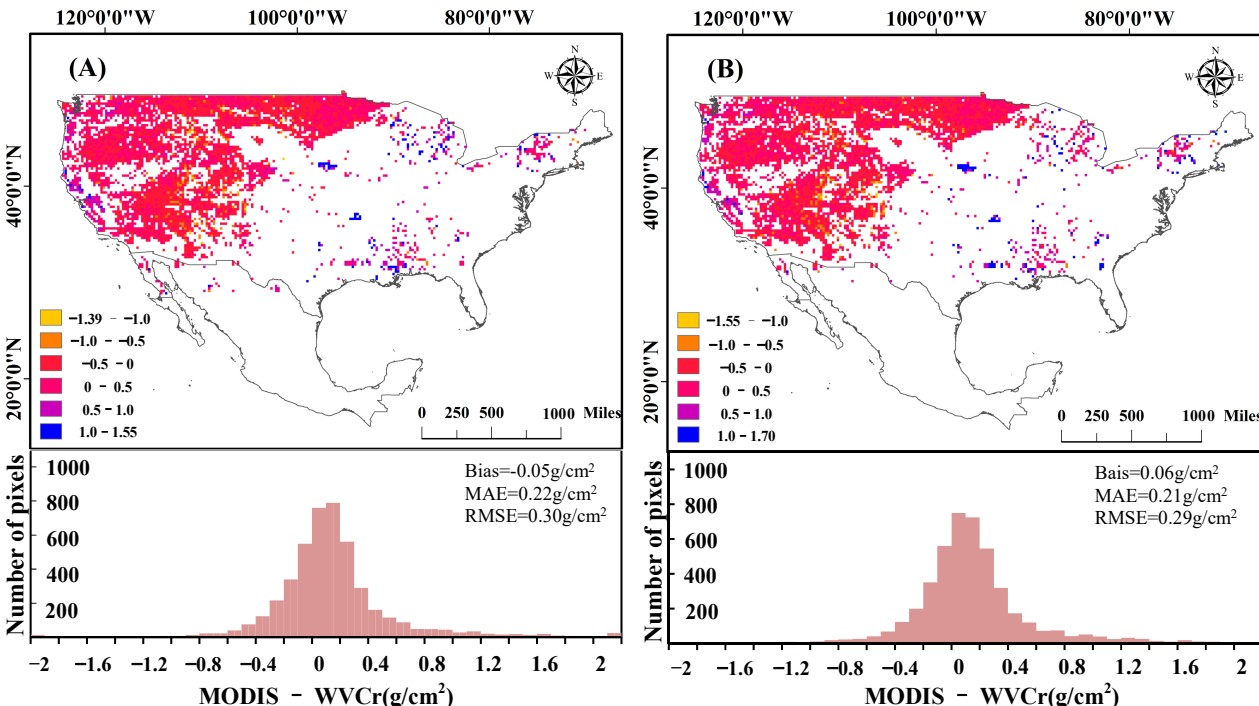

**Figure 9.** Differences (g/cm$^2$) between MODIS-VNIR products and WVC retrieved using PSP–DL method. (**A**) the difference between "MODIS-VNIR" and "4BTs", (**B**) the difference between "MODIS-VNIR" and "LST and LSE + 4BTs".

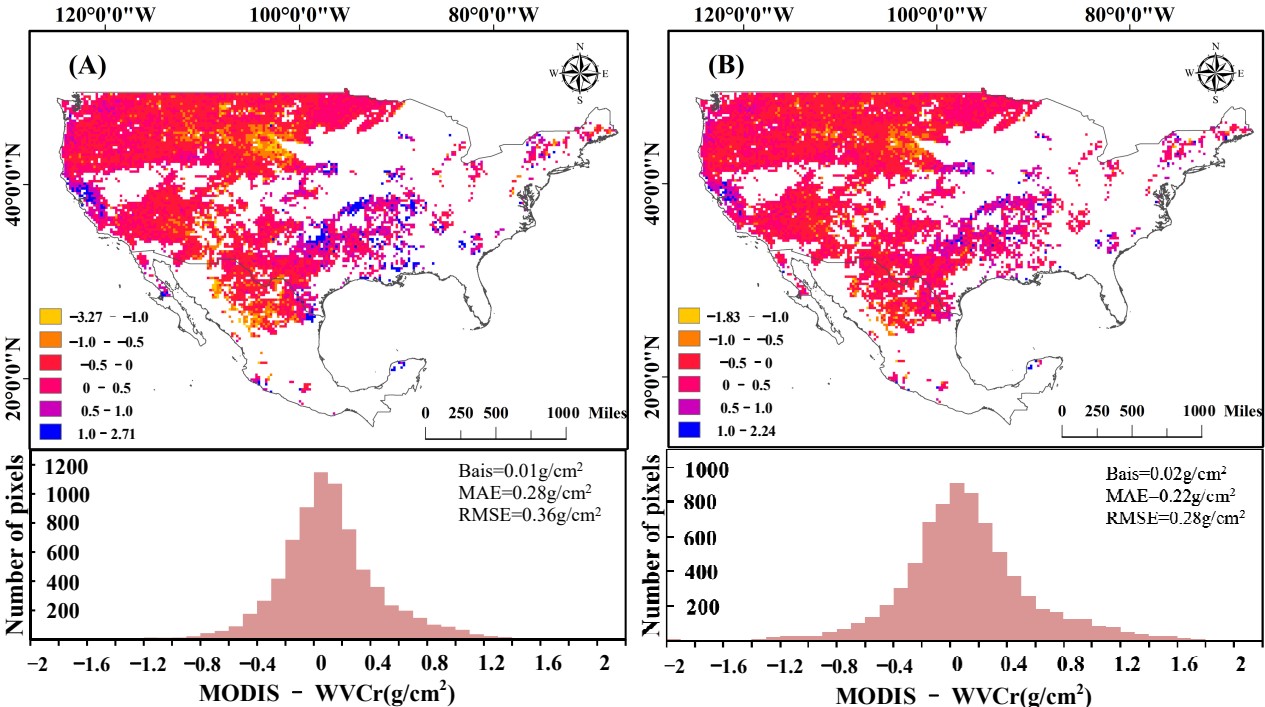

**Figure 10.** Differences (g/cm$^2$) between MODIS-TIR products and WVC retrieved using PSP–DL method. (**A**) the difference between "MODIS-TIR" and "4BTs", (**B**) the difference between "MODIS-TIR" and "LST and LSE + 4BTs". WVC$_r$ is the retrieval result of PSP–DL.

4.2.2. In Situ Validation

WVC estimates obtained by GPS are considered the most reliable [53], although there are temporal and spatial matching problems between remote sensing images and validation data. When the satellite transit time was closest to the GPS observation time, the relevant data was collected. In space, GPS is the point observation data of the surface, and there is a gap between the WVC within several kilometers corresponding to a pixel of a remote sensing image. To ensure data accuracy, GPS data were used only when the WVC values from the GPS observation points were close to ERA5 and MODIS data. We selected 255 observation points with a variety of surface types in southern North America, covering the period from January to December 2020, and extracted the pixels of the inversion corresponding to the latitude and longitude of the observation stations. As shown in Figure 11, the MAE of WVC retrieved by the combination of "LST and LSE + 4BTs" and ground synchronous data was 0.23 g/cm$^2$, with RMSE = 0.27 g/cm$^2$ and R$^2$ = 0.921. The MAE of WVC retrieved by combination of "4BTs" and ground synchronous data was 0.25 g/cm$^2$, with RMSE = 0.31 g/cm$^2$, and R$^2$ = 0.894. It can be seen from the analysis that adding prior knowledge did not significantly improve the retrieval accuracy, but the retrieval accuracy was more stable. Theoretically, the retrieval algorithm is more transplantable with ground prior knowledge. When there are bands that are very sensitive to the integrated water vapor content (such as bands 27 and 28), we can obtain relatively high retrieval accuracy without using LST and LSE as prior knowledge. If the satellite sensor has no TIR band that is very sensitive to the integrated water vapor content, it is more appropriate to use LST and LSE as prior knowledge to retrieve the integrated water vapor content.

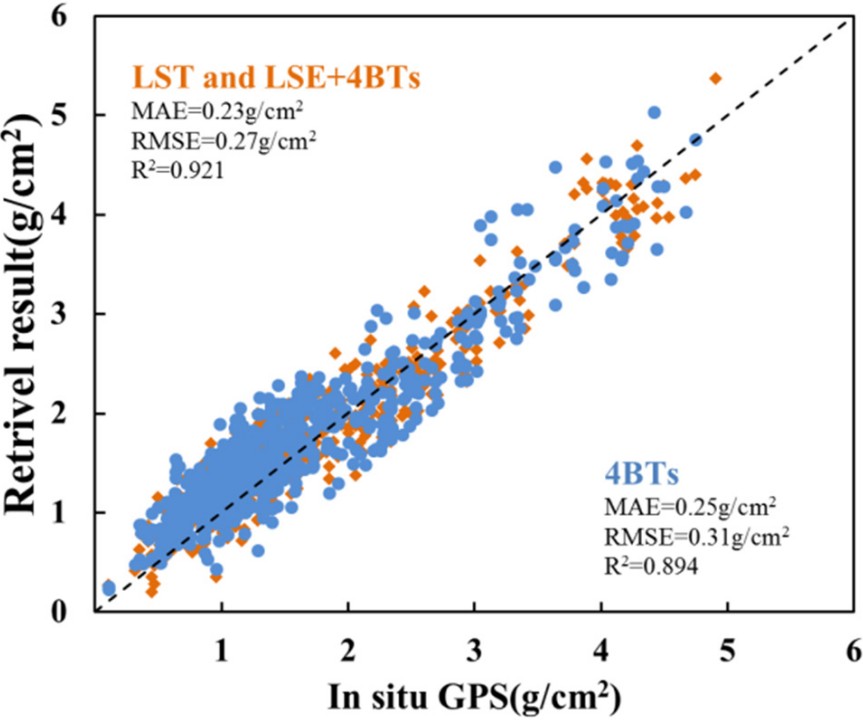

**Figure 11.** Comparison between WVC retrieval by PSP–DL method and GPS data.

5. Discussion

Sensitivity analysis is necessary in many applications where knowledge of probable WVC estimation due to possible errors in parameter determination is generally desired. In order to evaluate the impact of parameter errors on WVC retrieval, we need to perform a

sensitivity analysis for our algorithm. We utilized Equation (12) to compute the probable WVC estimation error as follows.

$$\Delta WVC = WVC(x + \Delta x) - WVC(x) \tag{12}$$

For our algorithm, the land surface radiation is the main influence factor for the accuracy of WVC. In this study, we utilized the MODTRAN simulation to analyze the sensitivity of LST and LSE. The retrieval results of WVC were obtained by changing the LST error (−1.5 K–1.5 K) and LSE error (−0.02–0.02). $\Delta WVC$ represents the difference between real WVC and retrieved WVC. In order to understand the sensitivity of prior knowledge to WVC, RMSE was selected as an evaluation index to analyze the changes of the two parameters. Figure 12a is the WVC error when we change the LST at the same step size (0.5 K). Figure 12b is the WVC error when we change the LSE at the same step size (0.005); here we made the sensitivity analysis for band 29, 31, 32 and change band 29/31/32 synchronously, with the same step size.

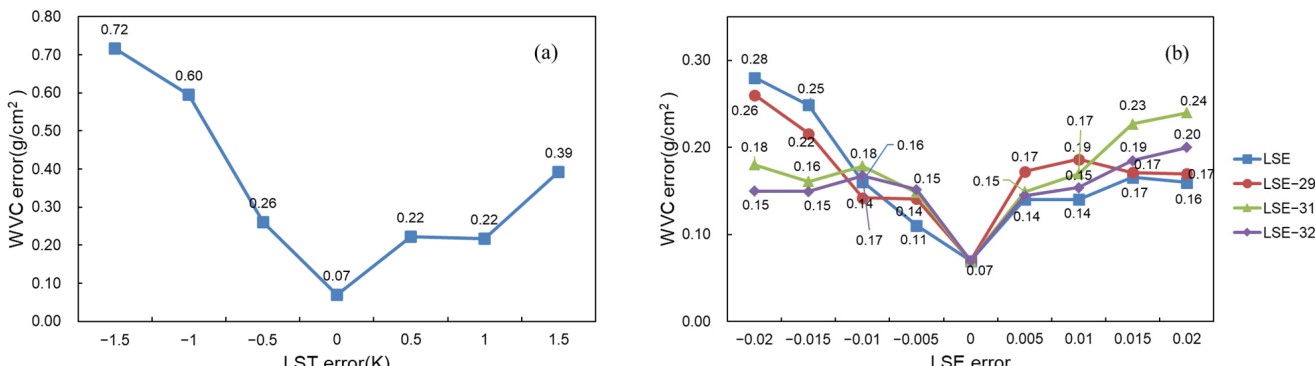

**Figure 12.** The sensitivity analysis of parameters. (**a**) change of WVC error with the change of LST error, (**b**) change of WVC error with the change of LSE error.

As shown from Figure 12 that the highest accuracy of the retrieval WVC is in the real LST or LSE, WVC error changes between 0.07–0.72 g/cm$^2$ when the LST error change is between −1.5 K to 1.5 K. Compared with the real LST, the larger the deviation, the larger the error of WVC retrieval. The error of WVC retrieval was largest (RMSE is about 0.72 g/cm$^2$) when the LST error was about −1.5 K, underestimating LST can lead to even greater errors, and the retrieval error of WVC increased suddenly when the absolute value of LST error was greater than 1 K. Figure 12b is WVC error when we change the band 29, 31, 32 emissivity (LSE) at the same error step size, and the error of WVC retrieval was the largest (RMSE is about 0.28 g/cm$^2$) when the LSE error was about −0.02. WVC error changed between 0.07–0.28 g/cm$^2$ when the LSE error changed between −0.02 to 0.02. The error of WVC retrieval was the largest (RMSE is about 0.28 g/cm$^2$) when the LSE error was about −0.02. Similarly, the larger the LSE deviation, the greater the WVC retrieval error, and underestimating LSE can lead to larger errors. In order to evaluate the sensitivity of the algorithm to LSE, we also analyzed the change of emissivity error of each band. When the emissivity of band 29 changed between −0.02 and 0.02, the WVC error changed between 0.07 and 0.26 g/cm$^2$. When the emissivity of band 31 changed between −0.02 and 0.02, the WVC error changed between 0.07 and 0.24 g/cm$^2$. When the emissivity of band 32 changed between −0.02 and 0.02, the WVC error changed between 0.07 and 0.20 g/cm$^2$. In general, it can be shown from Figure 12 that when the LST error was within ±1 K and the emissivity was within ±0.01, the retrieval error of WVC did not change much. The emissivity of the thermal infrared band has little change, so the algorithm is not very sensitive to an error of prior knowledge under the condition of large scale pixels. Deep learning coupled physics and statistical methods requires a certain amount of time in obtaining representative solutions from physical and statistical methods, as well as in deep learning training. However, once a deep learning model is trained, its speed is very

fast and it has significant advantages over traditional methods. The biggest advantage of this method is that we can establish different training and testing databases for different regions and seasons to further improve accuracy.

## 6. Conclusions

In this study, a new fully coupled paradigm is developed for robust retrieval of WVC from thermal infrared remote sensing data. The retrieval paradigm proposed by us is that a complete set of closed equations can be constructed between the input parameters and output parameters of deep learning in theory, which can be determined by physical logical reasoning through the radiation energy balance equation. If there is a strong correlation between input parameters and output parameters, deep learning can be directly used for retrieval. If there is a weak correlation between input parameters and output parameters, it is necessary to add prior knowledge to improve the retrieval accuracy of the output parameters. If we know a large number of representative solutions of the physical method, we can use deep learning to obtain the curve function of the solution through training. Physical model simulations provide us with the opportunity to obtain solutions for physical methods, so deep learning can replicate physical methods. Physical methods cannot describe all situations, and we can supplement solutions involving statistical methods with multi-source data.

The effectiveness of the PSP–DL was evaluated through simulations and experiments, and proven by obtaining satisfactory results. For the simulated data, the model with highest accuracy of the retrieved WVC yielded an MAE = 0.05 $g/cm^2$, RMSE = 0.07 $g/cm^2$, and $R^2$ = 0.993. Prior knowledge was found to improve the retrieval accuracy and reduce the number of required infrared bands. After comparing different band combinations, band 27, 28, 31, and 32 can obtain the optimal result, and it was also found that, if the band combination contains two bands that are very sensitive to the integrated water vapor content, high-precision retrieval can be achieved without prior knowledge. Reliable GPS site data were used for verification of the PSP–DL method, resulting in an MAE = 0.23 $g/cm^2$, RMSE = 0.27 $g/cm^2$, and $R^2$ = 0.921. The retrieval accuracy was slightly lower in very dry or wet environments, but this trend can be alleviated through the use of prior knowledge. In future applications, data availability, accuracy, and redundancy should be considered. Results from the case study of a part of North America showed that the WVC retrieved by the PSP–DL were consistent with the overall trends of two commonly-used WVC products.

**Author Contributions:** Methodology, Software, Validation, Formal analysis, Investigation, Data Curation, Writing—Original Draft, Writing—Review and Editing, R.M.; Conceptualization, Methodology, Software, Validation, Formal analysis, Investigation, Data Curation, Writing—Original Draft, Writing—Review, Editing, Project administration and Funding acquisition, K.M.; Software, Validation, Formal analysis, Investigation and Data Curation, J.S.; Resource, Formal analysis & Investigation, J.N. and S.M.B.; Resource and Editing, F.M.; Resource and Editing, G.D. All authors have read and agreed to the published version of the manuscript.

**Funding:** This project was supported by the Second Tibetan Plateau Scientific Expedition and Research Program (STEP)—"Dynamic monitoring and simulation of water cycle in Asian water tower area" (No. 2019QZKK0206), the National Key R&D Program of China (no. 2021YFD1500101), the Open Fund of the State Key Laboratory of Remote Sensing Science (no. OFSLRSS202201), Ningxia Science and Technology Department Flexible Introduction talent project (no. 2021RXTDLX14) and Fengyun Application Pioneering Project (no. FY-APP-2022.0205).

**Data Availability Statement:** Data openly available in a public repository.

**Acknowledgments:** The authors would like to thank the University Corporation for Atmospheric Research (UCAR) for providing Suominet Global Positioning System (GPS) site data, the NASA Earth Observing System Data and Information System for providing the MODIS data, and the ECMWF for providing the fifth-generation climate reanalysis data.

**Conflicts of Interest:** The authors declare no conflict of interest.

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
