# Peer review of "A Novel Physics-Statistical Coupled Paradigm for Retrieving Integrated Water Vapor Content Based on Artificial Intelligence"

_remotesensing, doi:10.3390/rs15174250_

Round 1

Reviewer 1 Report

 This study proposes a practical scheme for satellite remote sensing of water vapor amount, incorporating deep learning technique. In order to collect appropriate training data of deep learning, this scheme combines two types of method for preparing dataset: "physical method" that theoretically calculates radiances for a number of assumed water vapor contents with a radiative transfer model, and "statistical method" that collocates radiance data and water vapor data obtained from observation. The statistical method is expected to supplement the defect of the physical model. Validation experiments reveal that a band combination with four thermal infrared bands including very sensitive to the atmospheric water vapor are sufficient for retrieval of water vapor. Adding land surface temperature and emissivity as prior sometimes makes the retrieval results more stable.

Major comments

It is important for practical application of machine learning to prepare appropriate training dataset. This study provides useful information about training data collection for retrieval of atmospheric parameters from satellite remote sensing with applying AI. The purpose of this paper is therefore suitable to this journal. However, I think that details of the procedure of training data collection, which is most significant in this study, is not clear.

1. How far are separated between the distributions of collected data by physical and statistical methods? I think that if the distributions of radiance of the two dataset are similar, either can be eliminated. Or is the statistical method selective to pick up only data out of range of the physical method?

2. On the "physical method", description of radiative properties to be defined for radiative transfer calculation seems to be insufficient. I recognize that calculation of thermal infrared radiance by radiative transfer model needs to give atmospheric profile, especially of the temperature and water vapor content (g/cm3), although the range of water vapor path (0~4.5 g/cm2) is described. Explain details of the profile setting (or what profile models prepared in MODTRAN are used).  

3. Do values of the parameters correspond to regularly grid points of them (WVC, temperature, etc.) in a certain range of each parameter? If it is true, describe intervals of them and rough numbers of grids of each parameter. Or the values are randomly selected?

4. On the "statistical method", I have recognized from the explanation that empirical coefficients (∏ in Eq.10)  are first derived using MODIS radiances and WVC from GPS or so, then estimated WVC are provided as supervised data to carry out training of deep learning. If it is true, why are not the MODIS and WVC data used directly as the training data?

5. Is the number 25805 + 11803 (training data + test data) all of the statistical data? If it is not true, how to select training and test data from all collocated data (e.g., randomly)?

6. For both physical and statistical methods, how are the collected data divided into training and test?

Minor comments

7. The term "statistical method" seems to be inappropriate, because deep learning involves statistical models. How about "empirical method"?

8. The term "water vapor content (WVC)" generally means the density of water vapor (g/cm3), whereas WVC in the text corresponds to the water vapor amount integrated in the atmospheric vertical column (g/cm2). Replace it to water vapor path, integrated water vapor, or other usual terms.

9. The term "paradigm" used in the text might be inappropriate. How about "scheme"?

10. P2 L79, "ill-conditioned problem ~", What you want to mention here may be "ill-posed" (or "under determined")?

11. P8 L290, "the number of unknowns ~, at least four TIR bands are needed to construct the equations." I feel that this explanation is not logical. I consider Eq.6 as a result of an extremely simplified model, in order to manage to reduce the parameters of radiative transfer on the basis of several assumptions (e.g., ignoring vertical structure). Therefore, it may be reasonable that "Only these four variables can be determined at most from 4 (TIR) bands". Reconsider the explanation about determining the minimum number of input parameters required by DL.

12. P10 L342, describe the original of "PSP".

13. Figure 7, Make the color bar same range.

Reviewer 2 Report

 A novel physics-statistical coupled paradigm for retrieving water vapor content based on deep learning

Authors proposed a new retrieval method of atmospheric water vapor content based on a deep learning method. By combining retrieved data from a physical retrieval model and a statistical retrieval method, the more comprehensivetraining and test dataset were constructed. The author tried to use a deep learning method to integrate the advantages of physical and statistical methods to establish a more accurate retrieval product. Compared with existing operational retrieval products, it has also been proven that incorporating land surface temperature and land surface emissivity as prior information can indeed improve retrieval accuracy. However, the content of the paper is insufficient. In order to verify whether the new method can integrate the advantages of physical and statistical retrieval methods, authors should demonstrate retrieval results of the new method only relies on physical retrieval data or only relies on the data from the statistical retrieval method. In order to highlight the advantages of the new method, the article should further compare the new retrieval results with existing operational products. Given that these two aspects require further strengthening, major revision is recommended.

Line 32-33:“When land surface temperature and emissivity were taken as prior knowledge.”

These two kinds of data are very inaccurate. How can we ensure the accuracy of the prior information?

Line147-149:The retrieval paradigm of deep learning coupling physics and statistical methods proposed in this study is realized by composing the representative solutions of physical methods and statistical methods into training and testing database.”

Authors aim to merge the physical results and statistical products, what are the advantages of deep learning methods in this regard? If only the physical results or only the statistical products were used, would the results of the deep learning method be different? This should be clarified.

Line153-154:so as to make up for the defect that the forward model was difficult to simulate the mixed pixel.”

In fact, these two results may represent different physical relationships, can the deep learning method distinguish? Will mixing these two types of retrieval dataset lead to more confusion in the information received by the deep learning method?

Line 216-217: “but it cannot represent all the real situations, so it is necessary to supplement the solution of the statistical method.”

Why is the statistical method inaccurate in general, but accurate in those special cases?

Line 223-225: “The data will be selected only when the accuracy of the two atmospheric water vapor products is consistent.”

How do you quantitatively measure the consistency between the two kinds of dataset?

Figure7: “WVC retrieved from MODIS 481 bands 27, 28, 31 and 32 and MODIS LST and LSE products (MOD11).”

It is recommended to provide the spatial distribution and error characteristics of physical retrieval and statistical retrieval results, so as to better clarify the advantages of the new method.

Figure 10b: Is the significant system bias in the figure 10b caused by inaccurate LSTs and LSEs?

Line 510-512: “Comparative analysis showed that our algorithm has a high consistency with other products, and had certain advantages over those other products, such as situational adaptation through the supplementing of high-precision samples.”

This aspect should be the highlight of this study, but authors only provided a few sentences to talk about it. It is recommended to add 1-2 figures to demonstrate the advantages of the new product.

Line 521-523: “To ensure data accuracy, GPS data were used only when the WVC values from the GPS observation points were close to ERA5 and MODIS data.”

I don't understand. The accuracy of the data retrieved from the GPS observations should be higher than that of ERA5 data. Why is only data similar to the ERA5 data selected?

Round 2

Reviewer 1 Report

I appreciate your response to my comments. However, I think that it is necessary to add more explanation of the training data preparation, especially about setting of water vapor vertical profile for the physical method.

1. In Response 2, "The MODTRAN model is used to select the mid-latitude summer atmospheric reference model ~", the explanation for the used atmospheric model seems not to be appeared in the text. Describe obviously "the mid-latitude summer" in Section 3.1 or other.  
In addition, it is necessary to explain how to adjust the reference model to an arbitrary integrated WVC. Is the relative profile fixed and (absolute) water vapor density for all the altitudes multiplied by the same ratio ?

2. Are the impacts of the variation of water vapor profile estimated even if the integrated WVC is the same ? (e.g., the upper troposphere is humid (dry) but the lower troposphere is dry (humid) compared to the reference model. )

3. In Response 3, "We set change steps ~", describe the steps (intervals) of the values used in the radiative transfer simulation, especially for integrated WVC.

4. In Response 6, "We randomly divided ~", Does it mean that radiances corresponding to certain sets of the parameters in physical methods are not included in training data, because these belong to test data (and vice versa)? What is the influence of the "exclusion" of certain atmospheric conditions from the training data ?

Reviewer 2 Report

Unfortunately, I am unable to find the comparison between the effectiveness of the new machine learning method and traditional methods. I don't know which features in Figure 7-10 and Table 3-8 show this. So, it is still suggested that authors go further in this aspect, for example, is the new method being better for mixed pixels than physical retrieval methods? Or being better for clear sky pixels than statistical retrieval methods? Otherwise, we only need to use physical retrieval methods for clear sky pixels, and statistical methods for mixed pixels, where is the necessity of developing new methods?

Besides, you also should give some discussions about the advantages of results of the new method compared to the operational retrieval results? Is the new method fast? Does it need less auxiliary information? Or is it simply to verify the functionality of machine learning methods in this area?
